# Emotional, Behavioral, and Physical Health Consequences in Caregivers of Children with Cancer: A Network Analysis Differentiation in Mothers’ and Fathers’ Reactivity

**DOI:** 10.3390/cancers15133496

**Published:** 2023-07-04

**Authors:** Dorella Scarponi, Pierfrancesco Sarti, Veronica Rivi, Chiara Colliva, Elisa Marconi, Andrea Pession, Johanna M. C. Blom

**Affiliations:** 1Unità Operativa Pediatria Pession, IRSSC S. Orsola, 40138 Bologna, Italy; dorella.scarponi@aosp.bo.it; 2Department of Biomedical, Metabolic and Neural Sciences, University of Modena and Reggio Emilia, 41125 Modena, Italy; pierfrancesco.sarti@unimore.it (P.S.); veronica.rivi@unimore.it (V.R.); 3Local Health Unit of Modena, 41012 Carpi, Italy; chiara.colliva@unimore.it; 4Clinical Psychology Unit, Fondazione Policlinico Universitario Agostino Gemelli IRCCS, 00168 Rome, Italy; elisa.marconi@guest.policlinicogemelli.it; 5Radiation Oncology Unit, Department of Diagnostic Imaging, Radiation Oncology and Hematology, Fondazione Policlinico Universitario Agostino Gemelli IRCCS, 00168 Rome, Italy; 6Department of Medical and Surgical Sciences, University of Bologna, 40126 Bologna, Italy; andrea.pession@unibo.it; 7Centre for Neuroscience and Neurotechnology, University of Modena and Reggio Emilia, 41125 Modena, Italy

**Keywords:** parental distress, oncopediatric patients, graph theory, gender differences, anxiety, depression, somatization, hostility, risk factors, phenotype, behavior

## Abstract

**Simple Summary:**

This is the first study investigating the differences in psychological and physical distress between parents (i.e., mothers and fathers) of oncopediatric children using the Network analysis (NA). We also used the Kellner Symptom Questionnaire (SQ) for the first time with parents of cancer-diagnosed children. SQ data were analyzed using traditional statistical techniques (General Linear Model and ANOVA) and NA, an innovative technique which captures the complex interaction of core and environmental variables shaping the behavioral phenotype. Our concise yet effective measures provided a detailed understanding of distress levels, differentiating between psychological symptoms and well-being indicators for mothers and fathers.

**Abstract:**

Background: Pediatric cancer presents mental and physical challenges for patients and their caregivers. However, parental distress has been understudied despite its negative impact on quality of life, disability, and somatic disorders. Parents of oncopediatric patients experience high levels of suffering with their resilience tested throughout their children’s illness. Identifying at-risk parents and offering specific treatments is crucial and urgent to prevent or alleviate negative outcomes. Methods: This study used statistical and network analyses to examine symptom patterns assessed by the Kellner Symptom Questionnaire in 16 fathers and 23 mothers at different time points: diagnosis, treatment, and discharge. Results: The results indicated significantly higher distress levels in parents of oncopediatric children compared to the control reference population. Gender-specific differences in symptom profiles were observed at each time point, and symptoms showed a gradual but non-significant decrease over time. Conclusions: The network analysis yielded valuable insights that, when applied in clinical practice, can guide the implementation of timely treatments to prevent and manage parental distress, thus addressing long-term, stress-related issues in primary caregivers of children diagnosed and treated for cancer.

## 1. Introduction

Pediatric cancer is a complex disease that requires continuous care and therapy [1] and often necessitates hospitalization imposing mental and physical challenges to the (young) patients and their caregivers [2], on a personal, social, and professional level. Therefore, both oncopediatric patients and their caregivers are more susceptible to the exacerbation of emotional and physical disorders [1,3].

Although parental distress resulting from a child being diagnosed with and treated for cancer is associated with reduced quality of life, functional disability, and an increased risk of somatic disorders, this aspect has often been overlooked and/or under-studied [3].

Importantly, in a high proportion of parents of oncopediatric children, an association between the reported negative psychological effects and the time of diagnosis has been observed [4,5,6], while in a small subset of parents, effects diminish within the first few months following diagnosis; a larger proportion continues to report high levels of anxiety, depression, general psychological distress, and/or post-traumatic stress symptoms up to 10 years after diagnosis [4]. These long-lasting effects are thought to be due to the fact that—when faced with the diagnosis of their child’s cancer—parents quickly become immersed in the medical system that is managing the needs of their sick child, who—in turn—must cope with many challenges, such as side effects caused by treatment, life-threatening situations, and emotional problems [3]. While supportive and psychosocial care needs are a common parameter in studies regarding the oncologic pediatric patient in general [7], the specific psychosocial, emotional, physical, informational, financial, educational, and spiritual needs of their parents have been rarely investigated to date [8].

Therefore, studies aimed at analyzing and reducing the degree of distress in parents of pediatric patients are both necessary and urgent in order to allow healthcare professionals to identify, in a timelier manner, the individuals and families most at risk for the psychological and physical consequences of such a distressing and challenging situation [9].

To achieve this goal and promote a better quality of life for the family, it is essential to know and assess the parents’ needs, especially at the time of diagnosis, and continue to monitor these needs during the child’s therapeutic iter [10]. 

Moreover, to provide tailored preventive interventions and more individualized strategies, it is important to investigate the differences in distress between mothers and fathers of children with cancer, because if their way of expressing their distress is different, we will be better equipped to assist and intervene. Unfortunately, to date, very few studies investigated maternal and paternal differences in responding and coping with the stress associated with their children being diagnosed with cancer. This is mainly due to the fact that the statistical analyses, typically adopted in medical research, allow the analysis of the interaction of only a few variables at a time [11].

The role of parents in caring for their sick children is crucial, as they are directly involved in their children’s care and face many challenges and issues. Thus, it is important to identify the pressing needs of parents of sick children so that healthcare institutions can offer them comprehensive support as part of the care process. Psychologists have a fundamental role in managing young patients and their families and are often involved in assisting interdisciplinary cancer treatment teams, contributing to research, policy, and practice guidelines. Thus, in this complex scenario, psychologists are called to offer timely and effective interventions to improve the quality of life of parents of children diagnosed with cancer, to respond to their common unmet psychological and physical needs by paying attention to the physical and psychological difficulties they might display.

With these premises in mind, in this study, we tried to fill these gaps.

By recurring to PICO analysis as a standard method for the elaboration of clinical questions [12], in this study, we (1) defined the population and the problem to be investigated (i.e., emotional, behavioral, and physical health consequences in caregivers of children with cancer); (2) investigated the main differences in reacting to and coping with stress between mothers and fathers; (3) recurred to the Kellner Symptom Questionnaire to compare the distress levels in parents of oncopediatric children, compared to the control reference population; (4) defined—using the Network analysis—the gender-specific differences in symptom profiles starting from the time of diagnosis and up to three months into treatment; and (5) predicted how and in what time frame to implement personalized intervention to prevent and manage parental distress.

Thus, recognizing the importance of implementing interventions to prevent or address parental distress when their children are diagnosed with cancer, during treatment, and after discharge, we posed several key questions:(1)What are the specific areas in which parents’ experiences heightened vulnerability compared to the general population? How can these areas be effectively examined and studied?(2)Are there notable differences between mothers and fathers regarding their psychological responses to this highly stressful situation?(3)How do these risk factors vary over the course of time from cancer diagnosis to three months after diagnosis? Are there any discernible gender differences in the experience?(4)How can the findings of this study be expanded and applied on a larger scale to benefit a broader population?

The purpose of addressing these questions was to acquire a deeper understanding of the specific challenges encountered by parents of children with cancer. By doing so, we aimed to expose potential gender differences regarding mothers’ and fathers’ responses, examine how risk factors evolve throughout the journey over the first three months, and lay the groundwork for a wider application of the study’s findings.

Thus, by recurring to Kellner’s Symptom Questionnaire [13], we assessed the psychological symptoms (i.e., depression, anxiety, hostility, and somatization) and well-being indexes (i.e., contentment, relaxation, friendliness, and physical well-being) in parents of children diagnosed with cancers at different times (i.e., at the time of diagnosis and 1 and 3 months later). This highly sensitive clinimetric index has proven to yield clinical information that similar scales would fail to provide, and has a unique position among the patient-reported outcomes that are available [14,15,16,17,18]. To our knowledge, this is the first study to use this psychometric instrument, not with cancer patients themselves but with their parents. 

Furthermore, by adopting the network analysis, we investigated the maternal and parental phenotypes in coping with the stress associated with the diagnosis and treatment of cancer in their children. Thus, the data obtained from Kellner’s Symptom Questionnaire were analyzed using an integrated approach that combined classical statistical tools with network analysis. In this way, it was possible to study not only differences between parents of healthy (reference control) children and those of children suffering from cancer, but also to examine in a precise and predictive way if and how the psychological symptoms and well-being indexes vary between mothers and fathers of oncopediatric patients, over time.

By generating a network of interconnected variables, we were able to study the complexity of interactions between the questionnaire’s symptom scales with other variables of interest, as well as to compare patterns of gender- and time-point-specific parental phenotypes [19].

Specifically, the aims of this study can be summarized as follows: *The first aim* was to compare the levels of psychological and physical distress of parents at the time of diagnosis of the tumor of their child and investigate whether any differences existed compared to the Italian controls used in the validation of the Kellner’s Symptom Questionnaire. This allowed us to quantify the overall parental distress levels.*The second aim of this study* was to test the hypothesis that scores on the Kellner’s Symptom Questionnaire scales would change over time compared with the time of the child’s diagnosis. The time points for observing change were set at 1 and 3 months after diagnosis.*The third aim of this study* was to compare the scores of mothers and fathers at the various time points to analyze the presence/absence of gender differences in the evolution of psychological and physical distress from the diagnosis to three months into treatment.*The fourth aim of this study* was to implement an innovative Network Analysis allowing to study the relationships among the collected indexes and obtain a more comprehensive and complex view of the gender-specific symptom interaction of parental distress. The final endpoint was to individualize specific targets in order to direct psychological treatment and reduce negative outcomes.

## 2. Materials and Methods

### 2.1. Study Design

Parents of children diagnosed with cancer were invited to participate in the study to assess their needs. Inclusion criteria were: being parents of pediatric oncology patients at the time of diagnosis,initiation for the oncology patient of active inpatient/Day Hospital course of treatment,an expected course of treatment of at least 3 months at the facility where the study took place,being Italian-speaking or able to understand and speak Italian.

Exclusion criteria were:the presence of a previous oncological diagnosis of the patient in care,history of psychiatric diagnosis in one or both parents,inability to adequately understand the Italian language,refusal to participate in the study or non-filling of informed consent.

### 2.2. Participants

Parents were asked to participate in the study during their children’s hospitalization. Each invited person was informed about the purpose of the study. After obtaining informed written consent, subjects were asked to complete the questionnaire in paper version. 

Participants in the study were 23 parental, Italian, married couples of onco-Hematology pediatric patients. They were from central and southern Italy with schooling of eight years and above. 

Of these couples, the mothers had stopped working to handle and cope with their child’s diagnosis. None of the parents had previously been diagnosed with a psychiatric illness.

Of these pairs, 16 completed the three time-points at which the Kellner’s Symptom Questionnaire was administered. Of the other 7 parent pairs, only the mothers took the questionnaire; the fathers did not want to participate in the study.

The Statistical Power Test performed with a sample size of 16, maintaining a significance level of 0.05, and statistical power of 0.8 (a commonly accepted value), returns an r = 0.63. This suggests that despite the small sample size, significant effects of considerable magnitude (r > 0.5) can still be identified. The effects reported in the study, therefore, are robust and firmly established within the examined sample. It is worth noting that a larger sample size might have revealed additional effects, particularly regarding the differences between mothers and fathers and the three different time points.

### 2.3. Research Procedures

The test was administered to the parent, by a certified psychologist, according to the timeline described in the protocol. The first survey was administered at a time close to the oncological diagnosis, maximum within 7 days, the second survey was administered one month after the first, and the third survey was administered three months after the first. Each parent was provided with a restitution of their test results at a discussion interview with the certified psychologist who had administered the test. 

The study was approved by the Ethics Committee of Policlinico Sant’ Orsola IRCCS (code: CE 118-2017-O-sper), and participating parents signed informed consent to participate in the study. All participants were provided with information regarding the study’s objective and their research rights, particularly regarding the fact that there were no consequences if they decided not to participate. Participation in this study was voluntary.

### 2.4. Questionnaire 

*Symptom Questionnaire (SQ)* (Kellner 1987) [20] is a self-rated instrument for the assessment of psychological symptoms (depression, anxiety, hostility, and somatization) and well-being (contentment, relaxation, friendliness, and physical well-being). The SQ has been used in populations of adults, adolescents, and older individuals. In longitudinal studies and in controlled pharmacological and psychotherapy trials, it proved to be highly sensitive to changes in symptoms and well-being and discriminated between the effects of psychotropic drugs and placebo.

Findings indicate that the SQ has highly sensitive clinimetric indexes and represents one of the most sensitive Patient-reported outcomes (PROs) measure available. The SQ provides clinical information that other similar scales fail to provide and that may supplement the data derived from interview methods [14] (Appendix A).

### 2.5. Statistical Analysis

A General Linear Model was performed to analyze differences among groups, as previously described [21,22]. First, we analyzed our data for the assumption of normality using the Kolmogorov–Smirnov one-sample test for normality (K-S distance and P): all targets displayed a normal distribution. Moreover, preliminary analyses were carried out to test the homogeneity of variances between groups and independence. 

We processed the parents scores of the 8 subscales of the Kellner’s Symptom Questionnaire (i.e., Anxiety, Depression, Somatisation, Hostility, Relaxation, Contentedness, Physical Well-Being, and Friendliness) and compared with those of the reference sample (i.e., control group) used for the Italian validation of the test, in terms of average mean and standard deviation. It was not possible to perform one-way ANOVAs because the data matrices of the Italian validation article are not available. Repeated-measures ANOVAs followed by Tukey’s post hoc tests were used to analyze the results between T1, T2, and T3, and to divide them according to gender (i.e., mothers and fathers). Subsequently, individual comparisons between T1 and T3 were made with one-way ANOVAs followed by Tukey’s post hoc tests. All tests were defined as significant at *p* < 0.05. Data were presented as mean ± standard error (SEM). All statistical analyses were performed using SPSS v. 26.0 (IBM Corp., Armonk, NY, USA), while graphs were generated using GraphPad Prism v. 9.00e for Windows^®^ (GraphPad Software, Inc., La Jolla, CA, USA).

### 2.6. Network Analysis

Network analysis and all tables related to them were computed with R software (version 4.0.3/2020-10-10). Network models were estimated by separating the individual parents (father and mother) and the times at which data were collected (T1-T2-T3). Six different networks were generated and visually compared before being analyzed for differences in centrality, the significance of edges, and the predictability of variables.

The *mgm package* [23,24] was used for computing mixed graphical models (MGMs), as this software allows entry of both continuous and categorical variables. As all relationships (edges) represented in our models showed pairwise interactions (k = 2, interactions), the final networks reflected two-by-two variable interactions. 

Moreover, these relationships were controlled for all the other variables analyzed using the R software. The absence of a relationship between the two variables was indicative of their conditional independence given all the other variables. 

K-folds cross validation (10 folds) was applied to contain the number of false positives (Lq-penalization) and to choose the best tuning parameter in estimating network models, as reported by Jung et al. (2017) [25]. By recurring to this method, the dataset was randomly divided into 10 parts. Nine of them were kept in model training and the tenth part was tested. This procedure was then repeated 10 times, each time using a different “tenth” from the previous one [25].

For plotting the resulting networks, the *qgraph package* was used [26]. The final layout was determined with the Fruchterman–Reingold algorithm [27] by averaging the individual T1, T2, and T3 layouts among fathers and mothers of children patients diagnosed with cancer. In particular, the parental “average layout” was maintained over three times. 

Thus, the only variations were the edges between nodes. Importantly, the algorithm transforms the network into a system of particles having mass [28]. Therefore, nodes were interpreted as particles, whereas edges were as “forces” that push each other. The algorithm tried to minimize the energy of this physical system. A “uniform distribution of vertices” was then achieved. In addition to edges, what changes in result networks is the size of the nodes, which, in turn, represents the increase or decrease in the score (compared to T1) on the scale to which the variable (node) refers. 

Next, the predictability of each node in the network (i.e., nodes predictability) was calculated and represented. This measure is a representation of the amount of variance of each variable that is explained by the other variables with which it has connections [29]. In the case of continuous variables, the proportion of variance explained (i.e., R2) was used as a representative value. Thus, a value of 0 suggested that the node was not predicted by neighboring nodes (with which it had connections) in the network, while a value of 1 suggested that a node could not be perfectly predicted. For categorical variables, accuracy/correct classification (“CC”) was specified as predictability along with normalized accuracy (“nCC”) and model intercept (marginal) accuracy (“CCmarg”). Thus, by combining all these measures, it was possible to visualize, when necessary, the decomposition of total accuracy into “intercept” and “contribution of other variables.” Centrality analyses in the six networks were then calculated. 

Two measures were considered in the analysis: strength of centrality and betweenness [30,31,32]. The first one is described as the number of connections that a node has. Importantly, no reference to “incoming” or “outgoing” connections could be observed because the network model used was “undirected”. Therefore, when a variable in the network had many connections in a certain system, it could be considered as an influencing factor of the network. Furthermore, network edges had a weight when connecting two continuous variables, while they were binary (0–1) when connected continuous and dichotomous variables. Green edges represented positive partial correlations, while red ones represented negative partial correlations. Gray edges only indicated the presence of a not significant relationship. The thickness of an edge referred to the strength of that partial correlation/relationship. 

On the other hand, betweenness is an index of how well a node/variable fits into the ‘shortest paths’ between other nodes in the network and determines which nodes facilitate the system’s connections the most (Appendix A). 

We recurred to the algorithm that calculates the ‘shortest paths’, as previously described by Dijkstra (1959) [33], implemented in R and repurposed by Opsahl et al. (2010) [34]. Furthermore, in the interpretation of these indices, given the numerosity of the sample analyzed, bootstraps were performed to test their stability (bootstrapped strength centrality and bootstrapped betweenness). 

Finally, a clustering algorithm was executed to investigate the differences in connectivity patterns among the three groups from an alternative perspective. The clustering algorithm allowed us to identify a set of nodes that can be affected more quickly if one of them changes its state. Importantly, clusters are groups of nodes that are more interconnected within a larger network. We used the walk-trap algorithm, which measures similarities between nodes based on random walks across network connections (igraph package) [35], to capture the community or cluster structure in the graph [36,37]. The number of clusters identified was equal to the number of underlying factors in each dataset.

## 3. Results

In this study, we first analyzed the psycho-physical distress and well-being of the parents of children diagnosed with cancer and compared them with the reference sample (i.e., control group) used for the Italian validation of the test [13]. A total of 46 parents, consisting of 23 fathers and 23 mothers of Italian nationality, were enrolled. During the initial assessment (T1), all 46 participants completed the Symptoms Questionnaire. Among these couples, most mothers made the decision to resign from their jobs to be closer to their children. However, during the subsequent data collection at T2 and T3, seven fathers did not attend the hospital to complete the remaining assessments due to personal reasons. It is important to note that all participants had the right to withdraw from the study at any time, as stated in the informed consent. To ensure a fair comparison between the two groups, we excluded the corresponding seven mothers from the analysis, resulting in a final sample of sixteen pairs.

As shown in Figure 1A, in five of the eight subscales of Kellner’s Symptom Questionnaire (i.e., anxiety, depression, somatization, relaxation, and contentment), the scores of parents of oncopediatric patients were higher than those of the control group.

Interestingly, we found that the average score of the subscale ‘Anxiety’ at T1 also exceeded the values within the standard deviation of the control sample (4.25 ± 3.53). Unfortunately, it was not possible to perform statistical analyses between the results obtained from parents of oncopediatric patients and those of the control groups, as the use of the values extracted from Fava’s 1983 research did not allow access to the data matrix. However, these data suggest a highly stressful status in parents of children diagnosed with cancer (i.e., sub-clinical population). 

Furthermore, to highlight any differences in the parent’s reactions at different stages of the illness, we analyzed the trends of the symptoms over the three months following the diagnosis. 

Our working hypothesis was that there would be gradual reduction in the parent’s level of distress over time. Thus, to test our hypothesis we compared the mean scores of the four global scales of the SQ Test: Anxiety, Depression, Somatisation, and Hostility at the three time points.

Repeated-measures ANOVA and post hoc tests, conducted with the time factor at three levels (T1, T2, T3), revealed a significant effect only for the variable ‘anxiety’ in the reduction of the scale score between T1 and T3 (q = 3.85 *p* = 0.025). None of the other scales (i.e., depression, somatization, and hostility) showed a significant decrease in suffering (Figure 1B).

Next, we investigated whether gender differences (i.e., mothers and fathers) could influence the symptomatology outlined by the Symptom Questionnaire. Thus, we compared the scores obtained between mothers and fathers. To make the comparison reliable, only the parental pairs who completed all assessments three times were taken into consideration (i.e., 16 pairs, 32 parents in total). As shown in Figure 2, the trend of the scores obtained on all four scales of the Symptom Questionnaire of mothers and fathers were analyzed at T1, T2, and T3. We found that, following the diagnosis of cancer of their child, both parents showed high index scores and little influence of gender on the level of distress emerged. Then, we analyzed whether these differences were significant when comparing the measurement taken at T1 (i.e., when cancer was diagnosed in pediatric patients) and T3 (i.e., three months post-diagnosis). This comparison allowed us to observe significant differences between mothers and fathers at T3, related to the “Somatization” scale (One-way ANOVA, F = 6.86; *p* = 0.014).

Subsequently, network analysis was performed to map how the Symptom Questionnaire scales interacted with each other. This innovative approach made it possible to make important considerations from an interventional perspective that normal statistical analyses would not have permitted. An additional, dichotomous variable considered in the network was the sex of the child. The decrease/increase in the size of a node among the three time points reflected the percentage difference between the Symptom Questionnaire scale scores. The description of the results of the network analysis was performed by separating the three-time points and comparing the two parents (Figure 3).

At T1, we found an important role of Node 3 (i.e., Anxiety) in mothers, because many connections passed through this node (Betweenness = 1). These data suggest that the influence of this scale/symptom may have an important impact on all other nodes in the network. The anxiety component primarily acted on Nodes 4 (i.e., Depression), 5 (i.e., Somatisation), 6 (i.e., Hostility), and 7 (i.e., Relaxation). The partial correlations were positive. Thus, we concluded that a poor containment of anxiety may lead to more hostility, somatization, less relaxation, and an increase in the depressive component. On the other hand, in the fathers of children diagnosed with cancer, the most important nodes in the network were 7 (i.e., Relaxation) and 8 (i.e., Contentment), suggesting that at the time of diagnosis, interventions aimed at reducing the fathers’ anxiety may not be useful. In contrast, interventions promoting their sense of relaxation may be beneficial. Moreover, we found that Anxiety and Hostility were strongly connected.

Conversely, our data suggest that 1 month after cancer diagnosis, in mothers, anxiety lost importance in driving information within the network. Moreover, Node 7 (i.e., Contentment) moved to the foreground, which was greatly influenced by Node 5 (i.e., Somatization). Given these results, we investigated whether the sex of the child would play a role in the network. Interestingly, an important role of the sex of the patients in the network emerged, as suggested by the highest value of betweenness. 

In other words, depending on the gender of the patients, there were important interactions with the other variables. 

Additionally, Node 4 (i.e., Depression) became important in fathers, as this node had an extreme centrality and directed the information of the whole network with positive connections between individual nodes. Increases in this node led to greater impairment in all the other domains. In general, the size of mothers’ and fathers’ nodes decreased (i.e., we observed an overall decrease in scores on rating scales). The only score that was higher than T1 was Hostility in fathers, which, in turn, was directly influenced by Anxiety (Node 3).

Finally, we found that three months after their child was diagnosed with cancer, mothers and fathers displayed a decrease in the scores of all scales. Mothers’ improvements seem to be based on the interaction of symptoms on Node 7 (i.e., Relaxation) and secondarily on depressive symptomatology (i.e., Node 4). However, the network was still dense with connections, suggesting that a change in one node quickly influenced the other ones. 

Moreover, as all correlations were positive, our data suggest that an increase in anxiety, lack of relaxation, presence of somatization, or a downturn in mood are likely to, again, lead to a general worsening of the mother’s psychological condition. Alternatively, we found that in fathers, Node 3 (i.e., Anxiety) was the most important node in the network, having a direct effect on all other nodes of the Symptom Questionnaire. These data suggest that, by keeping the father’s anxiety under control, it may be possible to avoid the exacerbation of the other risk factors.

Finally, we performed cluster analyses (Figure 4). Interestingly, analyses suggest that the same set of variables (3-4-6-7) formed a cluster of their own only in mothers at both T1 and T3. In contrast, we found that, in fathers, although the clusters were more fragmented, the variables at each time point, that had the greatest level of betweenness, were always part of the largest clusters, suggesting that these variables may have a primary effect.

## 4. Discussion

To our knowledge, this is the first study investigating the differences in psychological and physical distress between parents (i.e., mothers and fathers) of children diagnosed with cancer using Network analysis. Furthermore, we adopted the Kellner Symptom Questionnaire for the first time in a subclinical population, that is, parents of children diagnosed with cancer. 

We demonstrated that parents of pediatric oncology patients, at the time of diagnosis, show higher levels of psychological and physical distress than the Italian control population extracted from Fava’s research (1983) [13]. 

Although the Symptoms Questionnaire was originally developed by Fava in 1983 (i.e., 40 years ago) [12], it has consistently demonstrated its effectiveness as a highly sensitive patient self-reported outcome measure [14]. In particular, its clinimetric properties have been well-established, including its ability to assess distress and levels of well-being, its high sensitivity even in studies with small sample sizes like the present one, its conciseness and simplicity, and its use of a yes/no response format that prevents patients from exaggerating or minimizing symptoms [14]. In the Italian National Oncology Plan (2023–2027) (www.salute.gov.it), the section dedicated to psycho-oncology acknowledges the substantial psychological and practical burden faced by caregivers. This burden is attributed to the significant amount of time dedicated to patient care (i.e., 40 h per week on average) and the various tasks involved, such as transportation, providing moral and psychological support, interacting with the care team, managing daily activities, assisting with medical prescriptions, and ensuring personal care and hygiene. Regrettably, there is a lack of sufficient focus on developing treatment plans specifically tailored for family members of patients. The emphasis is predominantly placed on addressing the needs of the affected individuals themselves. Consequently, our study employed statistical and network analyses as a methodological approach to identify the areas of the Kellner Symptom Questionnaire that demand greater attention in this realm.

Our data illustrate that some subscales of psychological and physical distress undergo significant modification at the time of diagnosis. These subscales include Anxiety, Depression, Somatization, Relaxation, and Contentment. Another important element to consider is that no difference emerged in the levels of Hostility between parents whose children were diagnosed with cancer and the control population. On the Anxiety scale, the scores obtained by the subjects belonging to the research sample exceeded even the standard deviation of the controls, reaching a level that, according to the Italian validation of the test, can be defined as critical. The psychological symptoms of parents who have just been informed of their child having cancer seem to be limited to more typically anxious expressions. Although no statistical analyses have been performed to determine the significance of these results, a higher level of distress was displayed by parents of oncopediatric patients compared to the Italian reference population. 

Next, we compared the scores obtained on the Kellner Symptom Questionnaire at three different time points during the first three months: at diagnosis (T1), after one month of hospitalization due to the first cycle of therapies (T2), and after precisely three months after diagnosis (T3). Mothers and fathers showed an overall gradual reduction in the level of distress. Attention should be paid to the anxiety scale, which reported the highest values at the first time point but showed a significant decrease at T3. These data can be indicative of the so-called “Adaptation to Illness” phenomenon [38]. In other words, it may be possible that, over time, parents may develop new psychological frameworks and a series of adaptive coping strategies following a traumatic event, such as the diagnosis of cancer in their children.

A central factor to be considered in the present study is that parents of the children who participated in this research had the opportunity to use various services offered by the Psychology, Psychopathology, and Psychotherapy Clinic of the Operating Unit within the department, both during hospitalization and Day Hospital treatment of their children. 

Thus, the decrease in the levels of distress and, consequently, improvement in the management of suffering was facilitated by the psychotherapeutic interventions timely provided, not only to the oncopediatric patients but also to their parents. These interventions included psychological and psychotherapeutic interviews and group play therapy. Moreover, support interviews, individual and family interviews, relaxation groups, psychotherapy, and therapies were provided, aimed to contain anxiety and other negative symptomatology, as well as enhancing the sense of self-awareness and self-efficacy.

Consistent with previous studies, we found that at the time of diagnosis, there were no significant differences in the management of emotional suffering between mothers and fathers [39]. However, parental phenotype differentiation emerged in the more advanced stages of the disease. In particular, when comparing the Symptom Questionnaire scores obtained from mothers and fathers at T1 and T3, we found that only in mothers was the somatization at T3 higher. This phenomenon can be explained by greater investment, on the part of the maternal figure, in the role of caring for the young oncopediatric patients. 

Alternatively, in mothers, the levels of somatization increased at the time of discharge from the hospital (i.e., T3), due to greater concerns caused by the lack of constant and present support from the healthcare staff.

Despite the relevance of the data obtained, the results from the statistical analyses did not allow us to perform personalized preventive interventions. 

By employing the network analysis, we were able to obtain new perspectives and valuable insights into the interplay of symptoms among parents, as captured by the variables in the Symptom Questionnaire. The network approach provided us with essential tools to interpret and understand the complex interaction of symptoms within the parent population.

This approach differs from the more commonly known social network analysis and represents the interaction between psychological variables which, in the field of psychology, is a relatively recent development, only implemented around a decade ago. In contrast, social network analysis originated in the 1930s and primarily served as a methodology for studying groups. In psychology, the prevailing perspective still revolves around the medical model of the “*Latent Disease Model*” used for psychiatric disorders, which posits the existence of an underlying cause that “generates” the disorder. However, network analysis, as described by Borsboom in “A Network Theory of Mental Disorders”, considers psychiatric disorders as outcomes arising from the interactions between symptoms, which are represented as nodes within a network that influence each other [40].

Hence, the innovative aspect of our study lies in considering parental burden as an interaction among various dimensions (such as depression, anxiety, hostility, ability to relax, etc.) that mutually influence one another, resulting in a distinct pattern of distress. Instead of merely attributing parental stress to their child’s cancer diagnosis, we have examined the multifaceted nature of parental burden through a single questionnaire. This approach enabled us to identify the predominant facet of distress experienced by each parent, providing valuable guidance to clinical psychologists in tailoring treatments accordingly.

In particular, we showed that it is not always the symptom with the highest score on the scale that is the most important in “driving” other symptoms and influencing them. This concept should always be considered when studying psychological suffering, as it is the result of a complex set of factors that are not often analyzed together [41]. 

Thus, as clinicians are called to provide the best possible treatment given the state of the art, network analysis has proven to be a valid “clinical” tool that could be included as a support and guide for the prevention and treatment of complex conditions, as they can model the interaction between affective and neurocognitive aspects intrinsic to the person, as well as demographic and environmental variables. 

Despite the interesting findings and innovative approaches of this study, future studies are required to address some limitations of the current research. 

## 5. Limitations

First, the experimental protocol, because of some common methodological difficulties in pediatric psycho-oncology research, such as the difficulty in establishing a fixed and well-defined setting or the need to respect medical timelines, might interfere with data collection. 

Furthermore, the fact that the study included an investigation into the experiences and psychological correlates of patients’ parents has made data recruitment more complex. There is a well-established practice, widely supported by the literature in psycho-oncology, of accompanying a patient’s medical treatment with psychological treatment. However, in the case of parents of oncopediatric children, there is sometimes greater resistance to addressing issues related to psychological distress. Research in this field, and other areas of healthcare practice, has made significant progress, but to define the problem more clearly, further experimental applications are needed alongside an already significant clinical effort. 

In addition, the small number of patients certainly led to less robust analyses but, particularly for networks, the bootstraps performed for network centrality indices were stable.

Due to the highly stressful circumstances faced by parents following their child’s diagnosis, it was not feasible to recruit and engage them in extensive interviews or additional testing. Therefore, a rapid screening method was preferred to at least capture insights into a range of emotions and moods to assist them better and more efficiently. Although there were other variables that could have made a significant contribution to the analyses, dividing the already small sample further based on additional demographic factors would have reduced the statistical sensitivity. Despite the limitations, the fact that significant effects were found in such a small sample sets the foundation for potential future extensions of this pilot study, involving more parents. Subsequent studies, then, could incorporate additional demographic variables and employ various tests to examine parental psycho-affective patterns in greater detail.

Also, the research project from which the study derives is still ongoing, and continuous administration of the questionnaire allows the collection of information that is now used by professionals in clinical practice to assist parents in managing their distress.

## 6. Future Practical Implications of the Study

In Italy, unfortunately, it is not yet common practice to comprehensively examine the components of parental distress when the child in the family is diagnosed with cancer, as the focus primarily remains on the child. However, we recently demonstrated the importance of exploring the interaction between psychosocial challenges and patient-related factors for understanding various disorders [5]. In particular, we found that different severe or chronic pediatric illnesses (including cancer) are characterized by disease-specific enhanced psychosocial risk and that risk is driven by disease-specific connectivity and interdependencies among various domains of psychosocial function. Although the present study did not aim to extensively analyze the psychosocial dynamics of the recruited families, it may serve as a stepping stone and provide evidence of what can be accomplished, even with a single questionnaire, to obtain a deeper understanding of parental distress. This knowledge can contribute to the development of more tailored psychological and psychotherapeutic interventions for individual cases. Future studies, however, should also incorporate a psychosocial assessment to provide insight into, for example, economic difficulties and the presence or absence of social support.

The findings from this Initial study serve as a starting point for a larger-scale multi-center study on a national level. The subsequent study will explore additional variables and aims to develop more targeted interventions that effectively mitigate the consequences of parental distress. 

By expanding the scope and depth of our present study in our future research efforts, we will be able to enhance our understanding and contribute to the development of more precise interventions to address the unique challenges faced by parents in the context of their child’s cancer diagnosis and treatment. The inclusion of parent-oriented care within pediatric oncology departments is crucial. Through the implementation of appropriate strategies, it will become possible to comprehend the specific needs of parents, respond to those needs in a positive manner, and provide effective care. By doing so, the aim is to minimize the needs of this vulnerable group and reduce the risks associated with unmet needs. Moreover, we aim to enhance the overall satisfaction and well-being of parents within the pediatric oncology setting. The ultimate goal will be to establish national guidelines for the personalized screening and treatment of parents with children diagnosed with cancer. The findings of this initial study lay the groundwork for a larger, nationwide multi-center study. This subsequent research endeavor will explore additional variables and aim to develop targeted interventions that effectively address the consequences of parental distress. By expanding the scope and depth of our current study in future research, we can enhance our understanding and contribute to the development of more precise interventions to meet the unique challenges faced by parents when their child is diagnosed with and undergoing treatment for cancer. It is crucial to incorporate parent-oriented care within pediatric oncology departments. By implementing appropriate screening and strategies, we can better comprehend the specific needs of parents, respond to those needs in a positive manner, and provide effective care. This approach aims to minimize the needs of this vulnerable group and reduce the associated risks of unmet needs. Furthermore, our goal is to improve the overall satisfaction and well-being of parents within the pediatric oncology setting. Ultimately, we strive to establish national guidelines for personalized screening and treatment of parents whose children have been diagnosed with cancer.

## 7. Conclusions

Psychological research in oncology has traditionally focused on patient suffering, but there has been a growing recognition of the challenges faced by caregivers in recent years. In pediatric settings, parents are typically the primary caregivers for their children with cancer, which puts them in a demanding position of accompanying and sustaining their child throughout the illness. This responsibility gives rise to a complex range of emotional experiences that can lead to significant psychological distress.

The objective of this study was to examine the intensity and characteristics of psychological distress in parents of children diagnosed with cancer. The findings underscore the importance of addressing the needs of parents from the early stages of the child’s illness.

In this study, the Kellner Symptom Questionnaire was validated as a reliable tool for identifying specific areas of distress, particularly anxiety symptoms, in parents of children diagnosed and treated for cancer. The results of the test–retest analysis indicated that anxiety is the most prominent symptom immediately following the diagnosis, although it may not be the primary concern for both parents in shaping their psycho-affective framework. Gender differences did not significantly impact the results during the initial measurements taken after diagnosis. However, over time, as children received treatment in the day hospital setting, mothers consistently reported significantly higher scores on the somatization scale compared to fathers. 

The use of innovative techniques, like network analysis, provided insights into the distinct interactions of symptoms experienced by the two parents. Although anxiety was the most prominent variable at the time of diagnosis for both parents, it was specifically identified as the leading symptom among mothers. These findings suggest that supportive treatments should be tailored and personalized, taking into consideration the individual experiences of each parent.

## Figures and Tables

**Figure 1 cancers-15-03496-f001:**
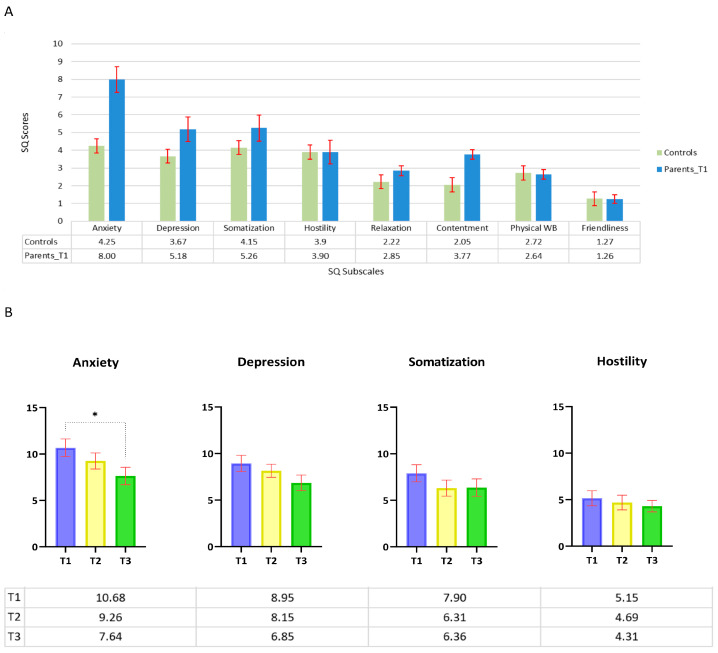
(**A**) Comparison of the scores from the administration of the QS to parents at T1 time and the averages corresponding to the Italian validation of the test. Below each pair of bars are the specific values. (**B**) Trend of the QS scores on the 4 global scales comparing T1, T2, and T3. Data are represented as means ± SEM and were analysed with two-way ANOVA followed by Tukey post hoc analyses. Significance differences are shown with * *p* < 0.005.

**Figure 2 cancers-15-03496-f002:**
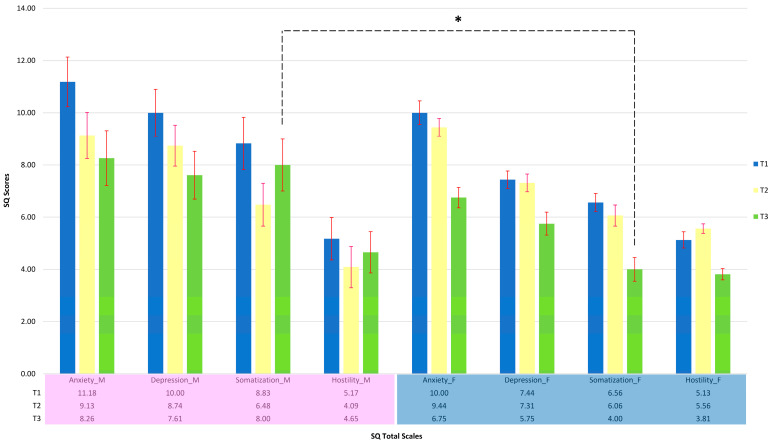
Mean scores with their error bar in the Symptom Questionnaire in the three measurements (T1, T2, T3). The pink box and the blue box at the base indicate mothers and fathers, respectively. Data are represented as means ± SEM and were analyzed with one-way ANOVA followed by Tukey post hoc analyses. Significance differences are shown with * *p* < 0.005.

**Figure 3 cancers-15-03496-f003:**
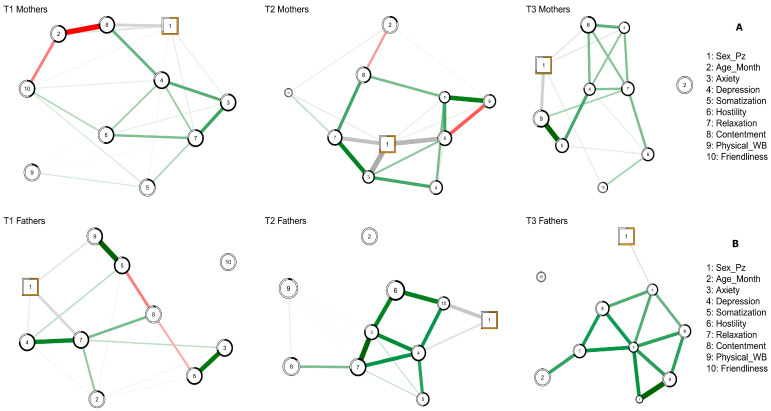
Maternal (**A**) and paternal (**B**) networks were ordered from left to right are the three-time points (T1, T2, T3). The circle nodes represent continuous variables, and the square one is the categorical variable. The black ring around each node represents the predictability value of that variable.

**Figure 4 cancers-15-03496-f004:**
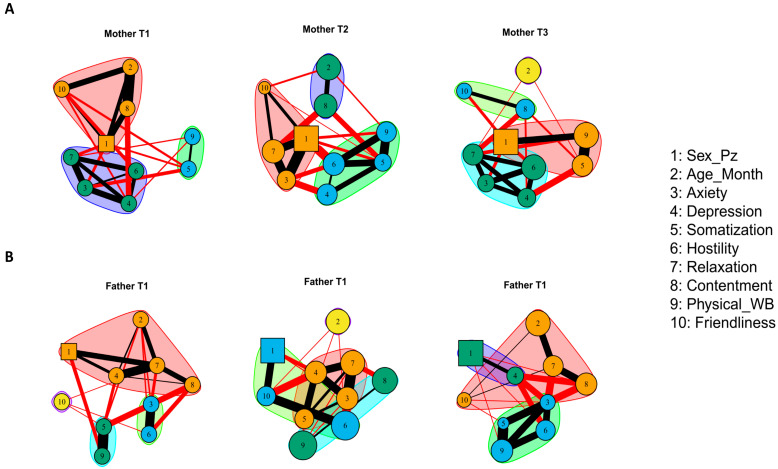
Networks of mothers and fathers put in “Average Layout” and with the clusters found through the *walktrap* algorithm. Part (**A**) of the figure shows the mothers’ networks, and Part (**B**) shows the fathers’ networks. T1, T2, and T3 follow one another from left to right.

## Data Availability

The data presented in this study is not publicly available due to privacy restrictions.

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
