# Peer review of "Emotional, Behavioral, and Physical Health Consequences in Caregivers of Children with Cancer: A Network Analysis Differentiation in Mothers’ and Fathers’ Reactivity"

_cancers, 2023, doi:10.3390/cancers15133496_

Round 1

Reviewer 1 Report

Thank you for the possibility to read and review your article about emotional, behavioural and physcial haelth consequences in caregivers of children with cancer. 

It is a well written article with clear aims, it gives insight in psychological and physical pattterns and may thereby attribute to the psychological care for parents of children with cancer. 

Some minor comments from my site:

In the introduction it not precisely clear from what professional background the autohors approach this topic. So I would suggest add some information about the role of psychologists in counseling parents of children with cancer.

Furthermore in the 'Symptome Questionnaire' is described with (only) one reference, while it is stated that the SQ has been used in many other studies. I would suggest to add some references here and preferabaly also a reference in which the Itealian SQ has been validated. 

I would suggest to start the results with a little bit more information about the parents. How many completed the questionnaires; what was the male-female distribution and what were the reasons for participants for 'not participating' in the study?

In the discussion the authors refers to the (40 yrs older) reference of Fava (1983) without taking into acount how the care for children with cancer (and their parents) have evolved over the recent years and how this possibly also have influenced the results and the comparisson between the studies. So add this to the discussion.

No specific comments

Reviewer 2 Report

Line 126: Was it clear that their needs would be assessed only and not addressed?

Line 131: Consider rephrasing as follows:
“an expected course of treatment of at least 3 months at the facility where the study took place”

Line 174: Please clarify PRO

Line 427-28: “they did not allow to perform”. I presume “they” refers to the statistical analyses.  Please insert “us” or another suitable agent between “allow” and “to perform.” 

Line 428-29: “recurring to” is an unclear phrase.  Please revise.

Thank you for inviting me to review this fascinating study.  The article seems clearly written for the most part.  I am unable to comment on the research methods, but they seem innovative, interesting, and thorough.  I have made a few editorial comments.

There are just a few instances of awkward phrasing.

Reviewer 3 Report

Overall, I found this to be an interesting study of an understudied topic in the cancer literature: parent caregiving distress within the pediatric cancer context. I believe that this study, despite a number of limitations, would be of interest to the readers. I have several suggestions that I believe would help to improve the quality of the manuscript.

First, I would like to see a stronger theoretical framework (or frameworks) that guide this study. For example, I can see how several theories from the stress and coping literature would be relevant (as well as theoretical work in the area of cancer caregiver burden). It would be helpful to have a theoretical framework that leads to specifically-stated hypotheses or research questions. The current manuscript does not list specific hypotheses or research questions, and the purpose and significance of conducting of certain analyses (especially the social network analysis section).

While the study measured states such as anxiety, depression, and hostility, the study does not account for what behaviors, perceptions, relational variables, etc. that are predictors of these states (which would be important to identify when developing an intervention study). In other words, are there relational problems between the mother and father, the parents and the child, with the oncologist, or other individuals within the family social network that may be contributing to these states. There are a host of other variables that could account for these as well (financial strain, uncertainty regarding the diagnosis, etc.).

We also know that social support plays an important role in mitigating the negative effects of anxiety, depression, etc. on health outcomes. However, the current study does not account for the social support structures (e.g., number of support providers, perceptions of the quality of social support, weaker ties vs. stronger support network ties, etc.) within the mother-father-child triad.

In short, there are a number of variables that influence the relationships between the variables chosen for the focus of the study that should either be accounted for (or arguments need to be presented for why they were not included in this particular study--as part of the rationale).

One of the key limitations of the study (which was noted by the authors) is the small sample size. However, it is understandable that it would be difficult to recruit the triads within this specific population. Of course, the few statistically significant findings in the results raises the question of whether or not the study was underpowered. 16 pairs is not a large sample size, and perhaps there was not enough power to detect significant effects. I feel that the authors do acknowledge this since they present other trends in the data, but I would like to see a deeper discussion of the power issue as a key limitation to the study.

The presentation of the social network analysis procedures is a bit cumbersome and confusing. I believe that this section could be reworked so that it provides a clearer picture of how the analysis differed from traditional social network analyses and a clearer explanation of how the key variables (nodes, etc.) should be interpreted in this case. Since most of the key variables in the study were measured self-perceptions, why not use SEM or a more straightforward analysis to parse out variance? In short, I am struggling to see what this "novel" approach is contributing to our understanding of pediatric cancer caregiver burden that other analyses could not provide.

Finally, I believe the manuscript would benefit from a deeper discussion of the theoretical and practical implications of the study findings for cancer caregivers. Again, I feel that if a stronger theoretical framework and clear hypotheses/RQs are presented in the front end of the manuscript, this will help set up the discussion section for a richer dive into the theoretical/practical implications of the findings.

Other than that, I thought the manuscript was well written and focuses on an important (and understudied) issue. There are a number of interesting findings, and I think the study makes a contribution to the literature overall.

I thought the manuscript was well written overall (no major issues).
